# Physics-Based X-ray CT Simulation for Synthetic Data Generation and Deep Learning Segmentation of Energy Materials

Nadia Mariana Leal Reyes*[1], Salvatore De Angelis[1], Estrid Buhl Naver[1,2], Luise Theil Kuhn[1], and Peter Stanley Jørgensen[1]

[1]DTU Energy, Technical University of Denmark.
[2]DTU Physics, Technical University of Denmark.
 {nmlre, sdea, ebna, luku, psjq}@dtu.dk

## 1    Introduction

X-ray imaging is an invaluable tool for non-destructive visualization of the internal structure and composition of energy materials such as battery electrodes and electrolysis cells. Energy devices are often composed of various granular, random microstructures mixed together. These devices and their components are studied to get information on their properties, morphology, and performance. However, data analysis is still largely performed with classical methods that rely heavily on manual input, making the process time-consuming and prone to human error. This limits the accuracy and throughput of microstructural quantification. Deep-learning approaches have shown strong performance in segmentation tasks [1], but their success depends on a large amount of training data, which are scarce in X-ray imaging of energy materials. Here, we present preliminary work towards being able to segment images of energy materials through the creation of simulated synthetic training data.

## 2    Simulation

Given that some of targeted real microstructures have a sphere-like shape, a preliminary structure generator was created in Python to produce images of non-overlapping random circles of different sizes. These images, representing a simple scenario of material sample slices, are then loaded into the simulation.

In X-ray Computed Tomography (CT), multiple X-ray projections are acquired at different rotation angles, each describing the beam attenuation along its path through the sample. These projections are reconstructed using the Filtered Back-Projection (FBP) algorithm to recover the spatial distribution of attenuation. The FBP method assumes a monochromatic X-ray beam, where each projection corresponds to attenuation at the same, fixed photon energy. However, in typical lab-CT scanners, the source is polychromatic, which breaks the monochromatic assumption and gives rise to artifacts known

---

*Corresponding Author.

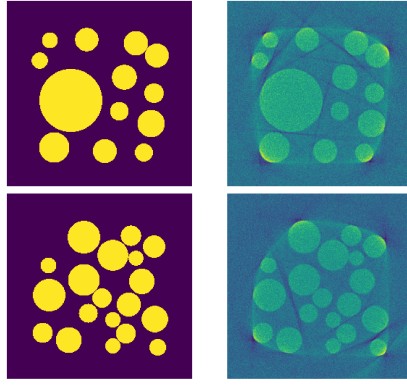

**Figure 1.** Left) Structures generated. Right) Polychromatic simulated reconstructions exhibiting strong beam hardening artifacts.

as beam hardening. An example of beam hardening artifacts can be seen in figure 1.

A simulation pipeline was implemented using the ASTRA Toolbox [2] and SpekPy [3] in Python. The source is simulated as a tungsten anode of thickness $12 \times 10^{-3}$m operated at 140 kV, yielding a mean photon energy of 38.51 keV. In the simulation, each projection is created by integration over the full source spectrum of photon energies, weighted by their relative intensities in the beam. As the beam traverses the sample, low-energy photons are absorbed more strongly than high-energy ones, producing nonlinear attenuation that leads to beam hardening—visible as shading and streak artifacts in the reconstruction. While such artifacts are commonly filtered during post-processing, they are retained here because they reflect the material's energy-dependent response. To mimic statistical and electronic noise, Gaussian noise with a mean of zero and a standard deviation of 0.01 is added to the reconstructed images. Figure 1 illustrates examples of procedurally generated input structures and their resulting simulated reconstruction.

## 3    Segmentation

Semantic image segmentation refers to the pixel-wise classification of an image into meaningful classes [4].

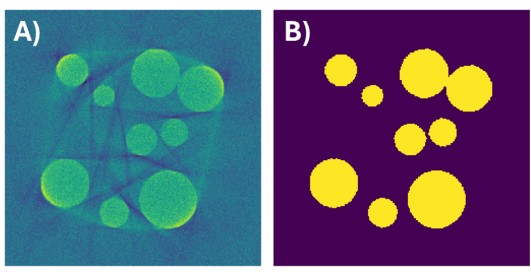

**Figure 2.** A)Grayscale tomographic simulation test image, normalized to unity. B)U-Net predicted class

In this context, it involves identifying and separating different material phases in X-ray CT images. Many deep learning models are developed to solve this task. Here, we use the U-Net [5], which has been widely employed and has shown excellent results [6].

The images obtained from the simulation pipeline are used as training data and the original images from the structure generator as the ground truth. The model receives as input 256×256 synthetic images, encodes them through three convolutional blocks (1→32→64→128 channels), and decodes them back to full resolution while merging encoder features via skip connections to produce a pixel-wise segmentation map of the same size.

The synthetic data consists of a set of 1000 images, divided into 70% for training, 20% for validation, and 10% for testing. The model was trained using Binary Cross-Entropy (BCE) loss function and Adam is used as the optimizer with a learning rate of 0.001 for 300 epochs. The model achieved a Dice score of 0.9943 and IoU of 0.9886 on the test set.

## 4 Results & discussion

Figure 3 shows an example segmentation of a test data image using the U-Net trained on synthetic dataset and the result of applying the Otsu thresholding method to the same image.

In this preliminary study, the U-Net captures the overall shape and position of the circular features. It provides a smoother and more accurate segmentation than Otsu thresholding. Most discrepancies appear along object boundaries. Figure 3 shows the difference between the class predicted by the U-Net and the ground truth (GT). Comparing Figures 3 C) and 2 B) highlights the deep-learning method's potential for accurate, automated X-ray CT segmentation.

## 5 Conclusion & Outlook

This work demonstrates the use of simulated physics-based synthetic X-ray data to train deep learning models for the segmentation of energy-material microstructures. Preliminary results with a U-Net

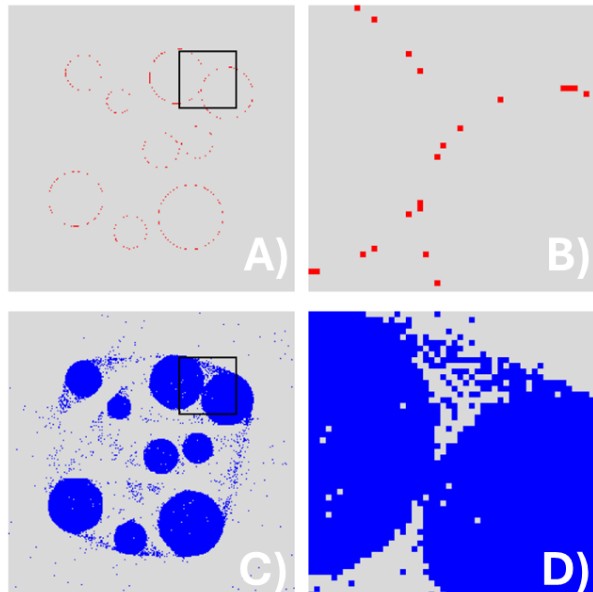

**Figure 3.** A)Difference between the U-Net predicted class and the GT. B)Magnified view of the region indicated in A. C)Otsu thresholding. D)Magnified view of the region indicated in C.

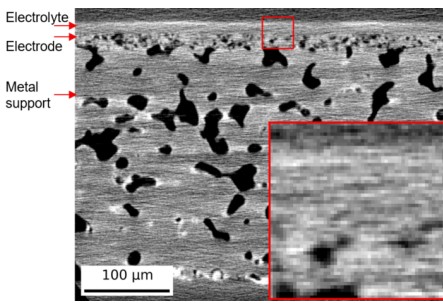

**Figure 4.** An X-ray µCT slice of a metal supported solid oxide cell is shown.

trained on simulated data show that the U-net can effectively learn to accurately segment microstructure in the presence of artifacts.

The project aims to develop a general segmentation framework for granular and random microstructures. Ongoing work includes extending the procedural microstructure generator to span a larger and more realistic variation of the microstructure and improving the simulation to produce realistic synthetic data for both laboratory- and synchrotron-based imaging. The next stage will focus on training with more complex synthetic structures and evaluating inference on real X-ray µCT data, such as the metal-supported solid oxide cell shown in Figure 4 that was taken with an Xradia Versa 520.

## 6 Acknowledgements

We acknowledge support from the Danish National Facility for Imaging with X-rays, DANFIX.

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
