# OpenReview forum: "Physics-Based X-ray CT Simulation for Synthetic Data Generation and Deep Learning Segmentation of Energy Materials"
_NLDL.org/2026/Abstracts_Track — NLDL 2026 Abstracts_

### Official Review · Reviewer_wc7W · 2025-10-28

**Soundness:** 3
**Correctness:** 3
**Rating:** 4
**Confidence:** 3

**Summary:**

This paper addresses the non-destructive visualization of energy materials using X-ray imaging. Traditional segmentation methods, which often rely on manual thresholding, are labor-intensive and require human intervention. The authors propose training a U-Net model on synthetically generated X-ray data to automate structure detection. Their approach outperforms Otsu thresholding, demonstrating the potential of deep learning for efficient and accurate segmentation in this domain.

**Strengths:**

The paper provides a clear and detailed explanation of the physics-based simulation used to generate synthetic training data. The experimental setup is thoroughly documented, ensuring reproducibility. Figures 1–3 are effectively selected to illustrate the problem and results, enabling readers to quickly grasp the key aspects of the work.

**Weaknesses:**

The paper lacks citations supporting the prevalence of classical methods in this field. While Otsu thresholding is used for comparison, it is not explicitly identified as a representative classical method in the domain. Figure 4, which illustrates a real-world application scenario, raises questions about the transferability of the synthetic data, as the synthetic images appear significantly simpler compared to the target real-world images.

---

### Official Review · Reviewer_a8gs · 2025-11-03

**Soundness:** 2
**Correctness:** 3
**Rating:** 4
**Confidence:** 5

**Summary:**

The paper presents a physics-based simulation framework for generating synthetic X-ray CT datasets of energy materials to enable deep learning-based segmentation without relying on scarce annotated real-world data. The authors simulate realistic polychromatic X-ray projections using the ASTRA Toolbox and SpekPy, modeling the tungsten anode X-ray source and beam-hardening effects. The synthetic data, derived from procedurally generated microstructures, are used to train a U-Net segmentation model. The model achieves high segmentation performance (Dice score: 0.9943, IoU: 0.9886) on test data, outperforming classical methods such as Otsu thresholding.

**Strengths:**

- The work addresses an important bottleneck in materials imaging: the lack of annotated training data for deep learning segmentation.

- The combination of physics-based simulation (ASTRA Toolbox + SpekPy) and data-driven segmentation (U-Net) is well-motivated.

- The reported Dice and IoU scores indicate that even with simple synthetic geometries, the U-Net can effectively learn to segment features and outperform classical methods.

- The authors explicitly plan to scale the framework toward more complex, realistic microstructures and real CT data, which suggests meaningful long-term impact and research potential.

**Weaknesses:**

- The current simulation is based on simple non-overlapping circular shapes, which poorly represent the complex, irregular, and multi-phase microstructures of actual energy materials. This limits generalization to real data.

- While the paper mentions planned experiments on real µCT data, no results are currently presented. The absence of domain transfer analysis makes it difficult to assess the real-world applicability of the proposed approach.

- The use of a vanilla U-Net with BCE loss, without exploring modern segmentation architectures or domain adaptation strategies, limits the methodological depth. Techniques like attention U-Nets, transformers, or physics-informed regularization could strengthen the contribution.

- The evaluation compares only against Otsu thresholding, which is a weak baseline. Additional baselines (e.g., classical machine learning, morphological segmentation, or learned models trained on limited real data) would better contextualize performance.

- Details on the computational cost of the simulation and training pipeline, as well as robustness to noise, spectrum variation, or parameter changes, are not discussed but are essential for practical adoption.

---

### Official Review · Reviewer_XVwR · 2025-11-03

**Soundness:** 3
**Correctness:** 3
**Rating:** 4
**Confidence:** 2

**Summary:**

The authors propose a physics-informed method to generate synthetic samples of X ray scans of energy materials. The goal is to use this to train deep learning segmentation to be used on such images. The method builds on domain knowledge of the process, starting by generating typically observed shapes to then introduce noise and common artifacts. A U-Net trained on the synthetic data is compared with Otsu-thresholding and shows promising results.

**Strengths:**

The approach of generating synthetic data through domain knowledge is interesting, and different from common approaches of generating synthetic samples based on the available sample data. The need for such methods is motivated, and it seems to perform well on the test case.

**Weaknesses:**

There is a lack of discussion on wether similar methods have been deployed before, e.g. physics informed methods to generate synthetic data, or other methods to generate synthetic data in this problem. Without knowledge of the specific dicipline (X ray for energy materials), it is also hard to evaluate if the used domain knowledge is reasonable, or if there are important aspects missing. Both these issues could be resolved by referring to related works, and are understandable issues in an abstract, but makes my evaluation less certain.

---

### Decision · Program_Chairs · 2025-11-05

**Decision:**

Accept

**Comment:**

The abstract is of interest to the community and should be presented at the conference.